# Biodegradation Behavior of Degradable Mulch with Poly (Butylene Adipate-co-Terephthalate) (PBAT) and Poly (Butylene Succinate) (PBS) in Simulation Marine Environment

**DOI:** 10.3390/polym14081515

**Published:** 2022-04-08

**Authors:** Bo Liu, Tonghui Guan, Gang Wu, Ye Fu, Yunxuan Weng

**Affiliations:** 1College of Chemistry and Materials Engineering, Beijing Technology and Business University, Beijing 100048, China; 2030401005@st.btbu.edu.cn (B.L.); 2030402045@st.btbu.edu.cn (T.G.); 2Collaborative Innovation Center for Eco-Friendly and Fire-Safety Polymeric Materials (MoE), National Engineering Laboratory of Eco-Friendly Polymeric Materials (Sichuan), State Key Laboratory of Polymer Materials Engineering, College of Chemistry, Sichuan University, Chengdu 610064, China; gangwu@scu.edu.cn

**Keywords:** PBAT, PBS, mulch, degradation, marine environment

## Abstract

Poly (butylene adipate-co-terephthalate) (PBAT) and poly (butylene succinate) (PBS) are polyester materials with excellent biodegradability under soil and compost conditions. However, the research on their degradation process in the marine environment is scarce. In this study, a more realistic simulation marine environment with sediment and marine organisms was developed, followed by investigation of the biodegradation behavior of PBAT and PBS mulch in it. The effect of aromatic structure, carboxyl end group content, molecular weight, and inorganic fillers on the degeneration of mulch was characterized by the changes in appearance, chemical structure, thermal properties, and crystallinity via Fourier transform infrared spectroscopy, differential scanning calorimetry, thermogravimetric analysis, gel permeation chromatography, element analysis, and X-ray photoelectron spectroscopy. The molecular weight of polyester blends decreased, while the content of the C-O bond in the composites increased, indicating that the samples indeed degraded. The degradation rate was measured with the CO_2_ release amount. The aliphatic polyester structure, lower molecular weight, higher carboxyl end group content, and the involvement of inorganic fillers facilitate the disintegration of polyester in the marine environment, which provides an effective method to construct materials with controllable biodegradable performance.

## 1. Introduction

Plastic pollution is considered a worldwide top environmental problem. Owing to their low-cost, good mechanical performance, and processability, petroleum-based plastics occupy a large portion of plastics used in our daily lives. However, the abandoned plastic waste could take centuries to degrade, which causes a great impact on the environment [1,2]. According to estimation, in 2020, 275 metric tons of plastic waste were produced, of which 4.8–12.7 metric tons entered the ocean. If there is no improvement in waste management infrastructure, by 2025, the cumulative amount of plastic waste that potentially enters the ocean from land will increase by an order of magnitude [3]. Moreover, when plastic is used in agricultural mulch and other applications, the plastic film residues in the soil environment are difficult to remove [4]. In recent years, biodegradable materials have been extensively researched to address these above-mentioned issues.

Biodegradable materials constitute a kind of material that can be decomposed into monomer particles in the natural environment, such as soil, compost, sea water, and other conditions for recycling or decomposing into small molecules, including carbon dioxide and water [5,6]. The degradation of biodegradable materials occurs under enzymatic or microbial actions, displaying little effect on the environment, and the utilization of biodegradable materials provides a promising strategy to palliate the environmental pollution problem caused by petroleum-based plastics [7,8,9]. Polybutylene adipate terephthalate (PBAT), a type of biodegradable aliphatic polyester, is widely applied in numerous fields on the basis of its good ductility and elongation at break, great heat resistance, and processing performance [10]. Especially, PBAT is a potentially effective choice as a degradable material used as agricultural mulch because of its fast degradation rate, high degradation rate, and less residue [11,12,13]. Nevertheless, the drastic defects in terms of the poor crystallinity and low melt strength of PBAT have a considerable influence on its application. As a result, many modification approaches, including chain extension and nanoparticle filling, have been performed to improve its comprehensive performances [2,14]. Polybutylene succinate (PBS), synthesized by polymerization reaction between 1,4-butanediol and succinic acid, is also a biocompatible and biodegradable aliphatic polyester with a more regular molecular structure than PBAT [15]. PBS has attracted considerable attention for its great biodegradable properties, processing performance, and mechanical properties [16,17].

To date, scientists have undertaken some research on the impact of the decomposition rate of bioplastic films in simulated soil, landfills, and industrial composting facilities. Due to the impossible complete recollection of biodegradable films during the application, it is indispensable to certify that the biodegradation of materials is not only the macroscopical disintegration or loss of weight, but also its complete remineralization or biomass conversion by microorganisms. It is verified that the decomposition rate of bioplastic films commonly depends on the chemical structure of the resin, the activity of the soil and compost, and the depth of burial of the sample [18,19,20,21]. The research on the biodegradation properties of degradable materials in aquatic environments, particularly in marine environments, is deficient [22,23,24]. The very limited knowledge and dramatically disparate experimental data on the biodegradation behavior of materials in marine call for the development of reliable standardized test methodologies and systematic research on the contributing factors of biodegradation properties. Meanwhile, field tests which have been conducted for several decades remain challenging [25]. It is absolutely imperative to establish an environmentally relevant simulation marine environment which can precisely reflect the realistic marine environment. Most of the plastic litter in the marine environment is in the coastal zone. The biodegradable polymers with higher density than sea water sink to the bottom of the sea after the surface of which is hydrolyzed and infiltrated with water. Therefore, the biodegradation of materials in the marine coastal zone can be modeled as aerobic biodegradation in the seawater/sediment interface (intertidal or sublittoral zone) [26].

In this work, a more realistic simulation marine environment with sediment and marine organisms was developed to reproduce the sublittoral habitat conditions of the coastal zone by a small-scale laboratory test method. The degradation behavior of PBAT and PBS with chain extenders and industrially produced biodegradable film products was systematically researched under the simulated circumstances of the real marine environment. The biodegradation behavior changes of film products with different bond saturation and film products obtained by adding different chain extenders were investigated, which made up for a deficiency in related fields and established a foundation for further research. Here, therefore, we investigated the polyester structure, molecular weight, carboxyl end group content, and the involvement of inorganic fillers to systematically research the degradation behaviors of PBAT and PBS, which could provide a feasible strategy for studying the degradation behaviors of other thermoplastic polyesters.

## 2. Experimental

### 2.1. Materials

All degradable mulch used in this work was obtained from Blue Ridge Tunhe Chemical Industry Co., LTD. (Xinjiang, China). The mulch made from poly (butylene adipate-co-terephthalate) with different carboxyl end group content is recorded as PBAT-26 and PBAT-10, which respectively contain 15.9 mol/t and 5.9 mol/t. PBS-27, PBS-15, and PBS-5 represent polybutylene succinate mulch with the different number-average molecular weight of 6.0 × 10^4^ g/mol, 5.5 × 10^4^ g/mol, and 4.9 × 10^4^ g/mol, respectively. THH and THB are from a biodegradable film formula with PBAT as the basic ingredient, auxiliary by polypropylene carbonate (PPC), and polylactic acid (PLA) at 5 wt% each. Different from THB, THH is made by using 5 wt% carbon black as filler instead of CaCO_3_.

The muddy sediment was collected beneath the low-water line at Ningbo, Zhejiang. The sediment was conserved at 4 °C and should be used within 4 weeks after sampling. The total organic carbon (TOC), pH, and nitrogen content of sediment are 0.77 mg/g, 8.0, and 0.16 mg/L, respectively. Artificial seawater with 34‰ practical salinity units (PSU) was used. Marine organisms, anemones, and clownfish were introduced in the simulation environment. Meanwhile, seaweed plants were not removed.

### 2.2. Biodegradation of Mulch in Simulation Marine Environment

The films were cut into 5 cm × 5 cm specimens for testing. The biodegradation testing of mulch of different formation was carried out respectively in the simulation marine environment tank at 30 °C. The composite samples were set flat on the sediment surface. One specimen of each sample was taken out every 15 days for testing.

### 2.3. Characterization

#### 2.3.1. Scanning Electron Microscopy (SEM)

The surface micromorphology of mulch specimens during the degradation were observed by SEM with the Quanta FEG (FEI, Eindhoven, Netherlands). SEM measurements were accomplished at accelerating voltages of 10 kV after the sample surface was sputtered with a homogeneous layer of gold.

#### 2.3.2. Differential Scanning Calorimetry (DSC)

The thermal properties and crystallization behaviors of PBAT and PBS degradation samples were obtained by DSC with the Model Q100 (TA Instruments, New Castle, DE, USA). Samples weighing 5–10 mg were quickly heated from room temperature to 180 °C, kept at a constant temperature for 3 min to eliminate the thermal history, then followed with cooling to −50 °C at a rate of 20 °C/min in nitrogen atmosphere. The reheating of samples was carried out at 180 °C at a rate of 20 °C/min. Before the analysis, all the samples were dried in vacuum at 60 °C overnight.

#### 2.3.3. Thermogravimetric Analysis (TGA)

A TA Universal V4.5A (TA Instruments, New Castle, DE, USA) was used to characterize the thermal decomposition mass loss of the samples from 40 to 500 °C with a heating rate of 20 °C/min under nitrogen atmosphere.

#### 2.3.4. Gel Permeation Chromatography (GPC)

The molecular weight of PBAT and PBS degradation samples was determined by GPC with the Waters 1515 Isocratic HPLC Pump (Waters, Milford, MA, USA). The analyses were performed with the eluent of dichloromethane at 40 °C.

#### 2.3.5. Fourier Transform Infrared Spectroscopy (FTIR)

The FT-IR spectra of PBAT and PBS degradation samples were obtained with a Nicolet iS50 FT-IR spectrometer equipped with Scientific iD7 attenuated total reflection adjunct (Thermos Fisher scientific, Waltham, MA, USA) in the wavenumber range of 4000–400 cm^−1^.

#### 2.3.6. X-ray Photoelectron Spectroscopy (XPS)

The chemical state of element on the film surface was investigated by X-ray photoelectron spectroscopy on a K-ALPHA^+^ (Thermos Fisher Scientific, Waltham, MA, USA) with an Al K α X-ray source (1486.68 eV photons). To compensate for surface charging effects, all binding energies (BEs) were referenced to the C 1s hydrocarbon peak at 284.8 eV.

#### 2.3.7. The Degradation Rate Measured with the CO_2_ Release Amount 

The carbon dioxide produced is continuously monitored in biometer flasks of 250 mL filled with 30 g marine sediment collected beneath the low-water line and 70 mL artificial seawater to determine the cumulative carbon dioxide production. The test films of 10 mg are perforated homogeneously and laid down on the surface of the sediment. The percentage biodegradation is given by the ratio of the carbon dioxide produced from the test material to the maximum theoretical amount of carbon dioxide that can be produced from the test material. The maximum theoretical amount of carbon dioxide produced is calculated from the measured total organic carbon content. The elemental composition of mulch organic compounds was performed using the Elementar vario EL cube analyzer (Elementar Analysensysteme GmbH, Langenselbold, Germany).

## 3. Results and Discussion

### 3.1. Micromorphology of Degraded Specimens

During the three-month-incubation, the film samples gradually became brittle and began to disintegrate. The specimens of PBS-5 and THH are so completely shattered that they could not be collected after 90 days. After 75 days of incubation, the micromorphology changes of the polymer film specimens are shown in Figure 1. With the increase of investment time, numerous corrosive cavities appear on the surface of materials, which caused by the permeate of water followed with the break of polymer chain. The surface of PBAT-26 with carboxyl end group content of 15.9 mol/t shows many severe cracks and holes, while PBAT-10 with 5.9 mol/t shows no other defects instead of a certain number of corrosive trenches. The reason for this is the random hydrolytic ester scission, the first stage of which, the degradation of polyesters, can be autocatalyzed by the carboxyl end group [14,27]. The surfaces of mulch film with inorganic fillers have numerous protruding bumps before degradation, many conspicuous cavities appeared in the surface of the blends after degradation. The easier infiltration of materials owing to the increased surface roughness with inorganic fillers promotes the hydrolysis of polyesters. More severe deterioration of the PBS film surface with lower molecular weight appears after the degradation, which is a result of the faster infiltration on account of the higher concentration of end group segments with more active segmental motion. Degradation occurred in mulch materials, the extent of which was affected by the saturation of polymer molecular structure and the existence of inorganic fillers.

### 3.2. The Changes of Thermal Properties and Aggregation Structure

The thermal stability and thermal decomposition of PBAT and PBS are evaluated by measuring the onset degradation temperature of 5% mass loss, peak degradation temperature, and ash content with TGA at a gradually rising temperature. The TGA curves of film samples before and after a 75-day biodegradation are shown in Figure 2. The onset degradation temperature of 5% mass loss and peak degradation temperature of film samples shift to lower temperature, which is due to the more irregular molecular structure including the lower molecular weight, wider molecular weight distribution, and larger end group content after the degradation. Although the existence of inorganic fillers in formulation can improve the material thermal stability, the PPC and PLA components with inferior thermal stability in THH and THB lead to the decrease of the onset degradation temperature of 5% mass loss and aggravate the deterioration of thermal stability after the incubation as well. The first stage degradation of incubated THH and THB occurs in the temperature range of 300–330 °C, the mass loss of which is mainly due to the complete decomposition of PPC. The mass loss of PLA from 330 °C to 370 °C appears in the TGA curve of THH after degradation for 75 days. This result indicates that the compatibility of blend components has decayed during the degradation. Furthermore, carbon black has a stronger effect on this compatibility decay than CaCO_3_ filler. The TGA curves of PBS reveal that this polyester shows weight loss of 5%, 50%, and 90% at 325, 400, and 424 °C, respectively, which is lower than that of PBAT. The decomposition temperature of PBS after degradation migrates to lower temperature not as significantly as PBAT with inorganic fillers, which can be attributed to the faster molecular fragmentation caused by the existence of fillers. It should be noted that this does not indicate that PBAT and PBS polymer chains are thermally stable at over 300 °C. These polyester materials show thermal degradation when extruded at 200 °C, which is implied by the decrease of the shear viscosity with prolonged time.

The amorphous regions in polyester are usually more susceptible to microorganisms and enzymes than the crystalline regions. The butylene adipate (BA) units and butylene terephthalate (BT) units in PBAT form a co-crystallization or a mixed-crystallization structure. The biodegradation and hydrolysis in the vulnerable aliphatic polyester units leads to a higher degree of structural regularity of the material during the degradation process and an elevation in the melting temperature, as shown in Figure 3. The melting enthalpy obtained by integrating the secondary melting peak area also gradually increases. It can be seen that the melting temperature of PBS gradually rises, which can also reflect that the amorphous regions are preferentially degraded. A small crystallization peak can be found before the melting peak of PBS, which is attributed to recrystallization [28].

The crystalline fraction $X_{c}$ (%) of PBAT and PBS mulches can be calculated by the equation:(1)$$X_{c}=\frac{\Delta H_{f}}{w_{P}\cdot \Delta H_{f}^{0}}\times 100\%$$
where $\Delta H_{f}$ is the enthalpy of fusion of the composite. $\Delta H_{f,\;PBAT}^{0}\;$= 114 J/g and $\Delta H_{f,\;PBS}^{0}\;$= 110.3 J/g are taken as the heat of fusion of an infinitely thick PBAT and PBS crystal, respectively. $w_{P}$ is the polymer content in the composite. The $X_{c}$ of PBAT and PBS in mulching films before and after degradation are listed in Table 1. The crystalline fraction of PBAT shows the direct impact of end group content and compound modification. The fast $X_{c}$ increase of PBAT with lower initial crystallinity indicates the rapid degradation of the irregular structure. The crystallinity of composites increases with the degradation time, which is a result of the faster biodegradation of amorphous regions by microorganisms and enzymes. 

### 3.3. Chemical Structure of Degraded Specimens 

The changes of chemical structure are the root cause of degradation behavior and properties. The infrared spectra of PBAT films with different degradation times are shown in Figure 4a. For the spectrum of PBAT-10, the peak in the region of 3500–3100 cm^−1^ can be attributed to -OH, which gradually enhances with the extension of the degradation time. The peaks at 2910 cm^−1^ and 2850 cm^−1^ can be attributed to symmetric and asymmetric stretching vibrations of C-H bonds, which gradually weaken with the extension of the degradation time. The peaks at 1260 cm^−1^ and 1245 cm^−1^ can be attributed to C-O bonds of aliphatic and aromatic, which tend to weaken gradually. These indicate that there is an intermolecular break in the sample structure. The peak at 1710 cm^−1^ with a shoulder peak at 1728 cm^−1^ can be ascribed to a low molecular weight ester and its free carbonyl group, which gradually weakens and widens during the degradation. The peaks at 1410 cm^−1^ and 1390 cm^−1^ corresponded to trans-CH_2_-plane bending vibrations. The carbonyl index (CI), the absorbance ratio of the carbonyl groups relative to the methylene peaks, can be achieved with the ratio of the absorbance at 1710 cm^−1^ to 1410 cm^−1^. The CI value expands from 4.0 to 4.9 after a 60-day degradation and subsequently decreases to 3.4 with the extension of incubation time to 90 days, indicating that the gradual ageing or degradation by oxidation is followed with the leaching of water-soluble hydrolysates. The relative peak strength of the C-O bond stretching vibration at 1102 cm^−1^ enhances with the prolongation of incubating time. This is the result of the formation of hydroxyl radicals by the breaking and hydrolyzation of ester bonds. The infrared spectrum of PBAT-26 shows the same changes as PBAT-10, which confirms the degradation of the materials. Meanwhile, the more significant changes of the relative peak strength of ester and hydroxyl radical in PBAT-26 further demonstrate the faster degradation rate of polyester with higher carboxyl end group content, which contributes better hydrophilicity and greater vulnerability to attack by water molecules.

Due to the addition of inorganic fillers, the characteristic peaks of CaCO_3_ and carbon black appear, while the more significant decrease of the relative peak intensity of ester around 1710 cm^−1^ emerges. This indicates the increased degradation rate caused by the existence of inorganic fillers, which introduced a plentiful supply of imperfect two-phase interface that can be attacked and hydrolyzed by microorganisms and hydrolytic enzymes.

In the spectra of PBS, as shown in Figure 4b, the peak at 1710 cm^−1^ is the stretching vibration of the carbonyl C=O bond while the peak at 1150 cm^−1^ is the stretching vibration of the C-O bond. The peak intensity of the C=O bond relative to the C-O bond decreases gradually during the degradation. The higher rate of change in relative peak intensity of the ester bond and hydroxyl radical indicates the more excellent biodegradable performance of PBS with lower molecular weight.

The changes of chemical compositions on the mulch film surface are the direct reflection of the degradation. According to the XPS analysis, the element content ratio of O/C shows a significant increase with the degradation. After 75 days of degradation, the increment of the O/C content ratio in PBAT-26 is 20%, while the increment of O/C content ratio in PBAT-10 is 21%, which indicates that the hydrolysis adequacy and rate of ester bond in PBAT with higher carboxyl end group content are both greater. The increment of O/C content ratio is 96% and 104% in PBAT doped with carbon black and CaCO_3_, respectively. The existence of inorganic fillers helps the degradation of polyester. The O/C content ratio in PBS changes little after degradation because of the adequate hydrolysis of the ester bond.

Figure 5 shows the C 1s core-level spectra of PBAT and PBS before and after 75 days of degradation. The C 1s core-level spectra can be curve-fitted into three peak components at BEs of 284.6, 286.3, and 288.6 eV, attributable to the C-C, C-O, and -COO- species, respectively. The peak component for-COO- is due to the ester linkages and the hydrolysate carboxyl in the biodegradable polyester. The peak component for C-O, meanwhile, is contributed by both the ester linkages and its hydrolysate. The hydrolytic degradation in blends owing to the cleavage of ester linkages and the reaction between water and the carbonyl groups forms the degradation products of hydroxyl- or carboxyl-terminated PBAT and PBS. The new peak of C=O at 287.7 eV in PBAT and PBS composites arises after the incubation in the simulation marine environment, which reveals the degradation of polymer films. The distinction between the characteristic peak intensity of PBAT before and after degradation is obvious, with the relative peak area proportion of C-O to -COO- increasing from 1.36 to 2.83 in PBAT-26 and from 1.54 to 3.55 in PBAT-10. Furthermore, the larger relative peak area of C=O in PBAT-10 demonstrates the higher degradation degree and rate of PBAT with more carboxyl end group. The relative peak area proportion of C-O to -COO- in THH is 2.79 and 3.36 before and after degradation for 75 days, while it is 1.91 and 4.42 in THB. The imperfect two-phase interface introduced in the polyester by the inorganic fillers promotes the degradation. After 75 days of degradation, the increase in relative peak area proportion of C-O to -COO- in PBS-27, PBS-15, and PBS-5 are 75%, 95%, and 112%, respectively. This result indicates that the degradation degree and rate of PBS-5 are the highest among the three formulations, which further demonstrates the negative correlation between the molecular weight and the degradation of polyester.

The molecular weight of polymer changes after incubation because of the molecular chain breakage or rearrangement, which is another significant change of material chemical structure during the degradation. The hydrolysis process is evaluated considering weight average molecular weight (*M_w_*) because it is less affected by low molecular weight than (*M_n_*). As shown in Table 2, the *M_w_* of PBAT and PBS generally decreases with the degradation time, indicating the scission of polyester molecule chain. The more significant polydispersity (PDI) growth of THH and THB than PBAT indicates that the molecular weight distribution increases, which proves that the addition of inorganic fillers promoted the degradation of PBAT. The promotion of carboxyl end groups on degradation is further demonstrated with the more dramatic reduction of *M_w_* of PBAT-26 than PBAT-10. It can be seen that there is a gradual increase of PBS molecular weight during the degradation. The maximum acceleration of PDI in PBS-5 degradation illustrates the negative correlation between the degradation degree of PBS and its molecular weight. 

### 3.4. Degradation Rate of PBAT and PBS Mulch Films

The carbon content of the film samples is determined with elemental analysis (EA). The theoretical amount of evolved carbon dioxide (ThCO_2_), the maximum theoretical amount of carbon dioxide evolved after completely oxidizing a chemical compound, is calculated from the carbon content and expressed as milligrams of carbon dioxide evolved per milligram of test compound, as shown in Table 3. The degradation rate is given by the ratio of the carbon dioxide produced from the test material to the ThCO_2_ that can be produced from the test material. The curves of degradation rate shown in Figure 6 indicate the occurrence of degradation. The carbon dioxide production rate of PBS film increases fast and gradually equilibrates after 150 days, which demonstrates that PBS can be hydrolyzed and further metabolized by microorganisms in the simulated marine environment. As a result of the more vulnerable ester bonds and larger amount of end groups in the polyester molecule, chains with lower molecular weight are attacked by water, followed with the oligomers by hydrolysis and enzymatic hydrolysis, which always begins from the end groups of macromolecules decompose into carbon dioxide and water by microbial metabolism. PBS-5 with the smallest molecular weight shows the highest degradation rate among these three PBS film samples. The degradation rate of PBAT film is much slower than that of PBS due to the increased spatial resistance conferred by the existence of benzene rings. PBAT-26 with higher carboxyl end group content and lower crystallinity demonstrates a higher degradation rate than PBAT-10. The carboxyl end groups in PBAT contribute better hydrophilicity and greater vulnerability of the ester bond to attack by water molecules. The carboxyl end groups work as the self-catalyzer in hydrolysis reactions. The amorphous regions in PBAT are usually more susceptible to microorganisms and enzymes, which results in faster degradation. Meanwhile, the PBAT film THH and THB show even higher degradation rates for the addition of other blend components (PPC and PLA) and inorganic fillers, which introduce a plentiful supply of imperfect two-phase interface that can be attacked and hydrolyzed by microorganisms and hydrolytic enzymes.

## 4. Conclusions

This study aimed to evaluate the degradation behavior of PBAT and PBS mulches in a more realistic simulation marine environment with sediment and marine organisms. The effects of the molecular structure of polyester, including the aromatic structure, carboxyl end groups, and molecular weight, as well as the polymer modification with inorganic fillers, was explored. The molecular weight of polyester blends decreased, while the hydroxyl content in the composites increased, demonstrating the molecular chain cleavage of PBAT and PBS during the degradation. The relative amount of C-O to -COO- in polyester molecules on the film surface was considered to analyze the degradability, and the significantly expanded ratio indicated the hydrolysis degree of the biodegradable polyester film. The degradation rates were obtained from the respiratory test, during which the complete biodegradation of polyester into CO_2_ occurred, and the degradation rate of PBS is higher than that of PBAT. The aliphatic polyester structure, lower molecular weight, higher carboxyl end group content, and the involvement of inorganic fillers facilitate the disintegration of polyester in the marine environment. This research provides an effective method to construct materials with controllable biodegradable performance.

## Figures and Tables

**Figure 1 polymers-14-01515-f001:**
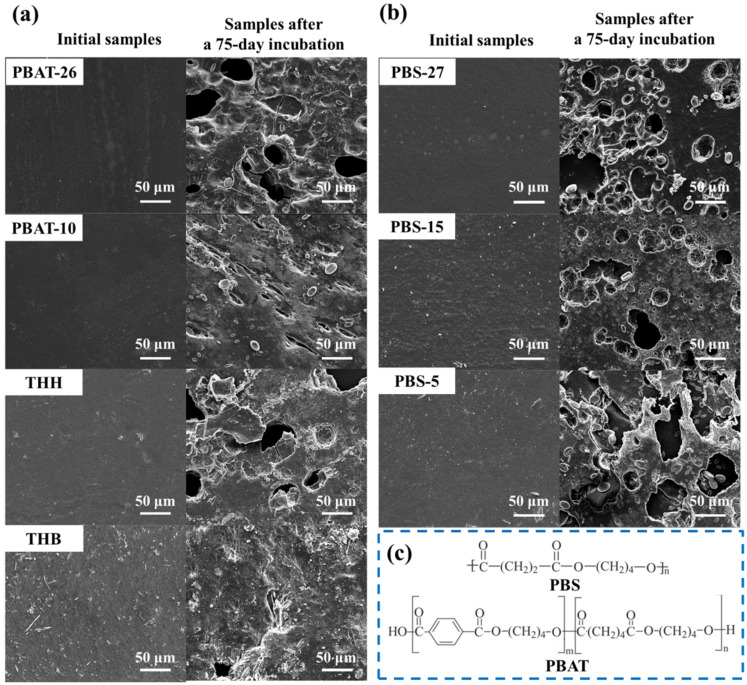
SEM images of (**a**) PBAT film with different carboxyl end group content (PBAT-26 and PBAT-10), PBAT film filled with carbon black (THH) and CaCO_3_ (THB) before and after a 75-day degradation; (**b**) PBS film with different molecular weight (PBS-27, PBS-15 and PBS-5) before and after a 75-day degradation. (**c**) Molecular structure of PBS and PBAT.

**Figure 2 polymers-14-01515-f002:**
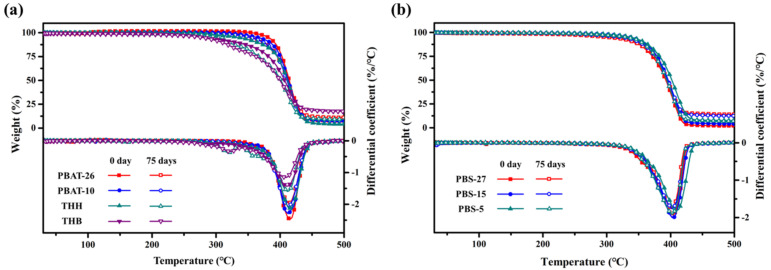
TGA curves of (**a**) PBAT and (**b**) PBS mulching films before and after a 75−day degradation.

**Figure 3 polymers-14-01515-f003:**
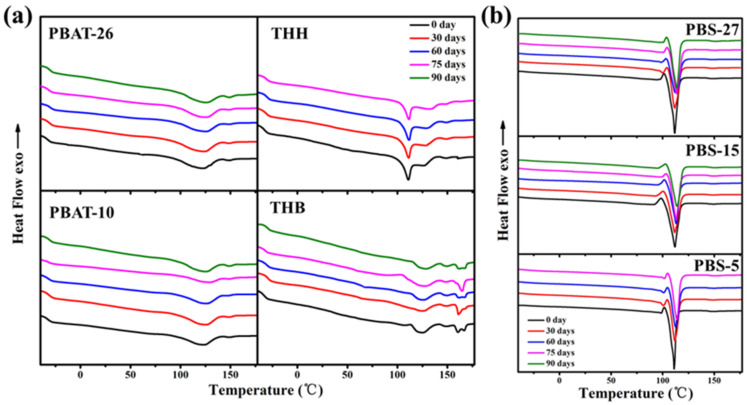
DSC melting traces of (**a**) PBAT and (**b**) PBS mulching films before and after a 75-day degradation.

**Figure 4 polymers-14-01515-f004:**
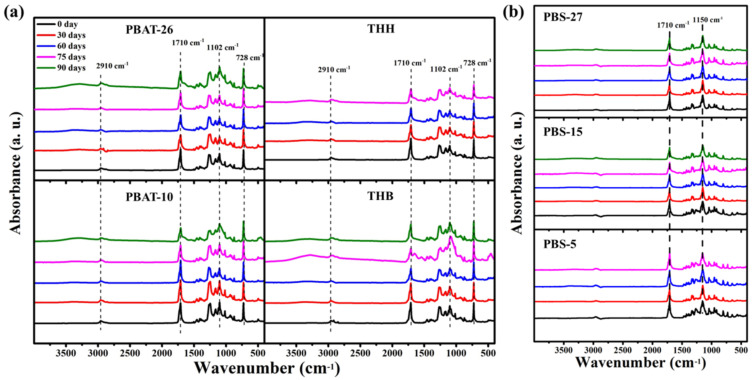
The FTIR spectra of (**a**) PBAT and (**b**) PBS mulching films before and after degradation.

**Figure 5 polymers-14-01515-f005:**
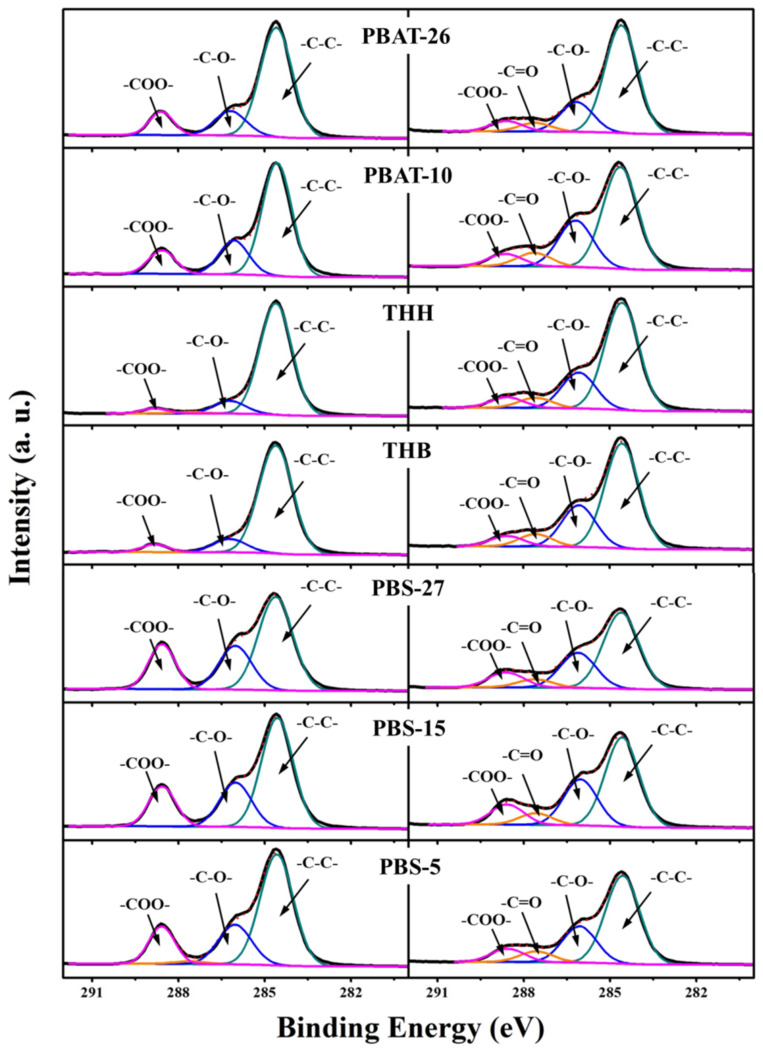
XPS C1s core–level spectra of PBAT and PBS mulching films before and after degradation for 75 days.

**Figure 6 polymers-14-01515-f006:**
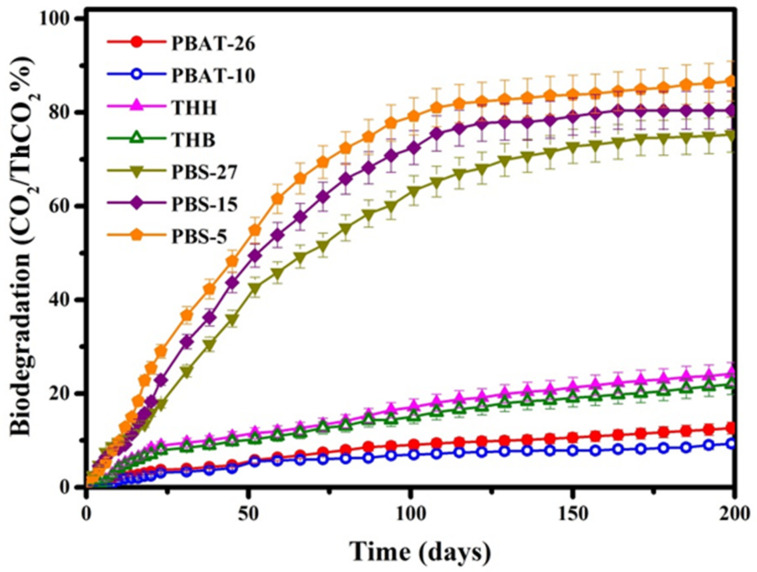
Degradation curves of PBAT and PBS.

**Table 1 polymers-14-01515-t001:** The $X_{c}$ (%) of PBAT and PBS in mulching films before and after degradation.

	0 Day	30 Days	60 Days	75 Days	90 Days
PBAT-26	11.8	13.4	17.2	18.9	20.7
PBAT-10	21.1	24.7	31.0	32.3	34.8
THH	10.4	14.1	14.5	20.3	
THB	9.7	11.5	14.3	14.7	16.6
PBS-27	1.7	6.3	6.6	9.0	14.8
PBS-15	7.2	8.1	8.3	10.1	12.8
PBS-5	12.7	13.0	13.3	15.5	

**Table 2 polymers-14-01515-t002:** The weight average molecular weight (*M_w_*) and polydispersity (PDI) of PBAT and PBS mulches.

		PBAT-26	PBAT-10	THH	THB	PBS-27	PBS-15	PBS-5
0 day	*M_w_* (KDa)	89	120	102	101	110	100	90
	PDI	1.6	1.6	1.6	1.5	1.8	1.8	1.8
60 days	*M_w_* (KDa)	88	117	97	98	103	94	88
	PDI	1.7	1.6	1.7	1.6	1.9	1.8	1.9
90 days	*M_w_* (KDa)	86	113	94	94	92	88	83
	PDI	1.8	1.7	1.7	1.7	1.9	2.0	1.9

**Table 3 polymers-14-01515-t003:** The carbon content and ThCO_2_ of PBAT and PBS mulches.

	PBAT-26	PBAT-10	THH	THB	PBS-27	PBS-15	PBS-5
C (%)	63.12	63.28	62.34	60.94	55.76	56.01	56.36
ThCO_2_ (mg/mg)	231.44	232.03	228.58	223.45	204.45	205.37	206.65

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
