# Peer review of "Biodegradation Behavior of Degradable Mulch with Poly (Butylene Adipate-co-Terephthalate) (PBAT) and Poly (Butylene Succinate) (PBS) in Simulation Marine Environment"

_polymers, 2022, doi:10.3390/polym14081515_

Round 1

Reviewer 1 Report

In the manuscript the authors investigated the degradation behavior in marine environment of butylene adipate-co-terephthalate and butylene succinate polyesters.

The topic is interesting due to the problem of environmental plastic pollution, so I believe the paper is worth publishing. However, the manuscript needs few minor revisions.

My specific suggestions:

1) section Introduction: 
Please describe the novelty and originality of this research in comparison with the literature. Please add an explanation about the similarities and differences between the results obtained in references [14-17] and the ones of this study.

2) In line with the publisher's recommendations, all acronyms and abbreviations should be defined the first time they appear in each of the sections: the abstract, the main text. Please complete. 

Author Response

Thank you very much for your suggestions. Based on the comments, we have made point-by-point amendments/ clarification to our manuscript. The revisions are described below. And we have indicated changes with red text in our manuscript for review except for many changes which are to reduce the grammar errors.

Comment 1:

section Introduction:

  • Please describe the novelty and originality of this research in comparison with the literature.
  • Please add an explanation about the similarities and differences between the results obtained in references [14-17] and the ones of this study.

Reply 1:

The introduction have been revised entirely to make it more lucid.

  • We have added the novelty of our research in the resubmitted manuscript.
  • We have changed the Ref [14] into a more appropriate one. We mentioned references [14-17] to explain ‘PBAT is a potentially effective choice as a degradable material used as agricultural mulch’ and ‘many modification approaches to improve PBAT comprehensive performances’.

Comment 2:

In line with the publisher's recommendations, all acronyms and abbreviations should be defined the first time they appear in each of the sections: the abstract, the main text. Please complete.

Reply 2:

Indeed, they should be. We have corrected in the resubmitted manuscript.

Reviewer 2 Report

The following issues must be addressed: 1. Introduction part should be improved in order to outline what is new and innovative in this work; 2. All the abbreviation should be explained; 3. Explain in more details why the there is such a difference in the sample surface degradation. 4. Statements such as “which dues to the decreased molecular weight after the degradation.” is too general, the authors should clearly give the explanation about the samples thermal behavior. 5. Why the authors consider that fillers induce faster molecular fragmentization? Explain in more details. 6. “The aggregation structure of PBAT and PBS film samples are illuminated with DSC analysis.” – reformulate. 7. Can you prove that the DSC peak can be attributed to PBS recrystallization? 8. The conclusion pars should contain the most significant results.

Author Response

Thank you very much for your suggestions. Based on the comments, we have made point-by-point amendments/ clarification to our manuscript. The revisions are described below. And we have indicated changes with red text in our manuscript for review except for many changes which are to reduce the grammar errors.

Comment 1:

Introduction part should be improved in order to outline what is new and innovative in this work.

Reply 1:

The introduction have been revised entirely to make it more lucid.

Comment 2:

All the abbreviation should be explained.

Reply 2:

Indeed, they should be. We have corrected in the resubmitted manuscript.

Comment 3:

Explain in more details why the there is such a difference in the sample surface degradation.

Reply 3:

The explanation of the differences in the surface degradation has been added in the resubmitted manuscript.

Comment 4:

Statements such as “which dues to the decreased molecular weight after the degradation.” is too general, the authors should clearly give the explanation about the samples thermal behavior.

Reply 4:

The explanation about the changes of thermal behavior has been enriched in the resubmitted manuscript.

Comment 5:

Why the authors consider that fillers induce faster molecular fragmentization? Explain in more details.

Reply 5:

We have made a mistake in this statement. The formulation of THH and THB is with fillers and polymer components of PPC and PLA. The decrease of onset degradation temperature is a synactic result. This has been clarifed in the revised manuscript.

Comment 6:

“The aggregation structure of PBAT and PBS film samples are illuminated with DSC analysis.” – reformulate.

Reply 6:

We have deleted this sentence to avoid misunderstanding.

Comment 7:

Can you prove that the DSC peak can be attributed to PBS recrystallization?

Reply 7:

Yes. The melting and recrystallization of PBS has been researched by Pof. Guo’s group via DSC. The relative result has been published. We have added this Ref in the resubmitted manuscript.

Comment 8:

The conclusion pars should contain the most significant results.

Reply 8:

The conclusion have been revised entirely to exhibit the most significant results.

Round 2

Reviewer 2 Report

The manuscript can be published in the present form.

Author Response

Thank you for your suggestion. We really appreciate the patient review and valuable comments.